physical chemistry/environmental science/ green chemistry

papermaking by-product, pulping process black liquor, activated carbons, surface area, pore volume, aqueous adsorption capacity

**Author for correspondence:**
Altaf H. Basta
e-mail: altaf_halim@yahoo.com,
altaf_basta2004@yahoo.com

This article has been edited by the Royal Society of Chemistry, including the commissioning, peer review process and editorial aspects up to the point of acceptance.

# Electiveness of agro-pulping process in the sustainable production of black liquor-based activated carbons

## Vivian F. Lotfy and Altaf H. Basta

Cellulose and Paper Department, National Research Centre, Dokki-12622 Cairo, Egypt

AHB, 0000-0003-1876-4378

During the production of paper pulp, the waste water loaded with organic materials from pulping process is discharged. Therefore, water treatment should be performed before disposing of such effluent. The use of such effluent for production of activated carbon will be effective in omitting the wastewater treatment and in obtaining the product required in many industries. In this respect, this paper deals with evaluating the performance of activated carbons (ACs) produced from black liquors (BLs) as by-products from three pulping processes of rice straw (RS) and sugar-cane bagasse (SCB), namely: alkaline, sulfite and neutral sulfite, which are coded SP, SSP and NSP, respectively. Elemental analysis and thermal analysis (TGA and DTGA) are carried out on the BLs, while the surface area ($S_{BET}$), micro-/ mesoporous distribution, adsorption capacity of methylene blue (MB) and iodine ($I_2$-value), as well as Fourier transform infrared spectra (FT-IR) and scanning electron micrograph (SEM) are studied on synthesizing ACs. The optimal pulping approach for achieving BL-based AC, with the following characteristics: specific surface area ($S_{BET}$) $\sim 921$ and $545\,m^2\,g^{-1}$, MB adsorption capacity 238 and $370\,mg\,g^{-1}$, and $I_2$-value 928 and $1255\,mg\,g^{-1}$ of BL-based ACs, are from neutral sulfite pulping of SCB (B-NSP) and RS (RS-NSP), respectively. These finding data are ascribed to the carbon content of BL, as well as greatest total volume ($V_T$ 0.786 and $0.701\,cm^3\,g^{-1}$) together with decreasing the volume of micropores/total (38 and 48%) of BL-NSP-ACs. It is interesting to note that the AC provided from RS-NSP has greater adsorption capacity for $I_2$ and MB than the AC produced from RS-pulp fibres.

## 1. Introduction

Based on environment and sustainability demands, great efforts were made by the researchers in the last decades on finding the green approaches to upgrade the industrial wastes in the

production of valuable products. The production of paper sheets from available natural wood and agricultural sources triggers interesting activities in related industries [1,2]. One of the major technical approaches for paper pulping agro-wastes is based on using the black liquor (BL) by-products. [1]. The conventional approach for the use of BL is through thermal process for recovering chemicals [3], but unfortunately the recovery process, especially from alkaline pulping of rice straw (RS), is extremely difficult and it demands greatest energy [4,5]. The trials to overcome the environmental impacts from disposing such wastes were focused on enhancing the application of this waste. The importance of lignin was first highlighted in 1977, via introducing salt lignosulfonate or soda lignin as a sacrificial agent [6–9], for resins, enhancing the strength of concretes, water absorption inhibitor and fluidization agents, for synthesizing chemicals (e.g. resorcinols, quinones, vanillin), paints, agricultural as controlled release of agrochemicals, dispersing and stabilizing agents for dyes and ceramic suspensions, as well as source of carbon materials [10–18].

Functionalized materials, e.g. cellulose derivatives and activated carbons (ACs) have received attention in the last decades due to their great applications in many fields, especially for chelating metals and removing the organic and inorganic pollutants from waste water [19–23]. The ACs are regarded as one of the famous porous carbon materials and produced from natural or synthetic precursors, e.g. coal, biomass and organic xerogels through subjecting these materials to carbonization as well as physical or chemical activation processes. The process of preparation plays a profound effect on the surface properties and porous structure [24–28]. These porous materials have wide applications, e.g. purification, electro and low-toxicity wood products [20,29–31]. With regard to the use of the black liquors (BLs) of paper pulping process, most of the literature focused on isolating the lignin, by precipitation via acid from black liquors, which were disposed from soda and Kraft processes [14,32,33]. But, for economic purposes, using whole black liquors is still rare [34]. In the present paper the black liquors that resulted from pulping of RS are used for the production of ACs. The types of pulping processes of RS (alkaline, acidic and neutral) are optimized and compared with BLs produced from sugar-cane bagasse (SCB). Moreover, the performance of BL-based ACs will be compared with previously produced ACs from RS and SCB pulp fibres [35]. The choice of SCB for comparison is based on this residue already being used in local paper mills for production of paper products.

# 2. Experimental procedure

## 2.1. Materials

*AC-Precursors*: two available Egyptian agriculture wastes (RS and SCB) were used as agro fibres for the pulping process to get the black liquors (BLs), which was further applied for the production of ACs. Three different reagents were used for the pulping process, namely, sodium hydroxide, sodium sulfite and a mixture of sodium sulfite and sodium carbonate (neutral pulping). The black liquors (BLs) from these pulping processes were coded as BL-SP, BL-SSP and BL-NSP, respectively.

*Activating agent*: Phosphoric acid, 85% was supplied from Farbwerke Hoechst AG (Germany), and *Aqueous adsorbates*: MB dye and Iodine were purchased from Alfa Chemicals Co. and El Nasr Pharmaceutical Chemical Co, respectively, for estimating the adsorption efficiency of investigated ACs in aqueous phases.

The experimental work is summarized in Figure 1.

## 2.2. Black liquors and their activated carbon preparation

The work in this article is focused on studying the efficiency of the activated carbon produced from the BLs, which resulted from the pulping processes of RS and SCB. The conditions of the pulping process and its BL-coding were as follows: (I) alkaline pulping using sodium hydroxide that is equivalent to 6.55% $Na_2O$ [BL-RS-SP] [BL-B-SP], (II) acidic pulping using sodium sulfite that is equivalent to 6.55% $Na_2O$ [BL-RS-SSP] [BL-B-SSP] and finally (III) neutral pulping using mixture of sodium sulfite and sodium carbonate with a mass ratio 4 : 1 that is equivalent to 6.55% $Na_2O$ [BL-RS-NSP] [BL-B-NSP]. All the pulping processes were applied in autoclaves with liquor ratio 5 : 1 at temperature 140°C for 2 h. The pH of the resulted BLs is adjusted to 6 and dried in oven at 60°C.

The dried BLs are activated using the phosphoric acid with a ratio 3 : 1 (BL-powder : phosphoric acid), followed by pyrolysis in horizontal tubular furnace at 450°C for 60 min [36], in the absence of air. Finally, the carbons were washed with distilled water till they became neutral and dried in oven at 105°C.

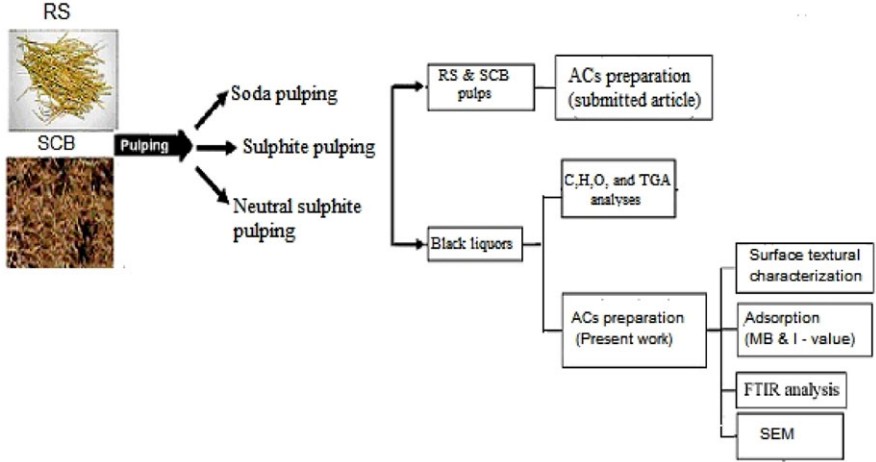

**Figure 1.** Experimental work.

## 2.3. Characterization of BLs and BL-based ACs

### 2.3.1. Black liquor characterization

— *Elemental analyses*: Vario El Elementar (Germany) elemental analyzer in the National Research Center-Egypt was used to determine the main elements (C, N, H and S) of the resulted BLs, while the oxygen contents are obtained by differences.

— *Thermal analysis* (*TGA*): Thermo-gravimetric analyses (TGA and DTG) of the BLs used ACs preparation were done using Perkin–Elmer Thermal Analysis Controller AC7/DX TGA7. The analysis was performed with a heating rate of $10°C\,min^{-1}$ and nitrogen flow rate of $50\,cc\,min^{-1}$, under non-isothermal conditions. This study was carried out to follow the weight loss of BL versus carbonization temperature and to estimate the final weight remains due to total inorganic elements included in the BL.

### 2.3.2. Activated carbon characterization

#### 2.3.2.1. FTIR spectra analysis
Infrared spectra were recorded with a Jasco FT/IR, Nicolet and Model 670. The samples were mixed with KBr and pressed as tablets. The bands were recorded in the region from 4000 to $400\,cm^{-1}$ with deuterated triglycine sulfate detector. This test characterized the functional group of the prepared BL-ACs.

#### 2.3.2.2. Scanning electron microscopy
The morphology of the investigated ACs is examined by scanning electron microscopy (SEM). The samples were exposed (gold coating, Edwards Sputter Coater, UK) using a quanta FEG250 system running at 20 kV. The samples pictures were magnified to $40\,000\times$.

#### 2.3.2.3. Surface textural characterization
Textural characterization of activated carbon samples prepared from the resulting BLs is carried out by nitrogen adsorption–desorption isotherms that are performed at $-196°C$ (77 K) using Nova (USA) 3200 Nitrogen Physisorption Quantachrome Instrument. The samples are degassed in oven at $250°C$ for 24 h. This test was performed to get the surface area of ACs (BET), total pore volume, particle size distribution and $t$-plot.

#### 2.3.2.4. Iodine value
Iodine number, accepted as the most fundamental parameter, was used to characterize activated carbon performance. It was measured according to the procedure established by ASTM [37].

Calculations involved in iodine value estimation are as follows:

$$\text{iodine value} = C \times \text{conversion factor } (mg\,g^{-1})$$

$$\text{coversion factor} = \frac{40 \times \text{Mol wt. of iodine } (127) \times \text{iodine normality}}{\text{carbon wt.} \times \text{blank reading}}$$

$$C = \text{blank reading} - \text{volume of sod. thiosulfate consumed after adsorption.}$$

## 2.3.2.5. Batch equilibrium and kinetic studies of MB adsorption

Six different concentrations from 100 to 600 mg l$^{-1}$ of MB solution are added to BL-ACs powder with liquor ratio 400 (0.025 g/10 ml). The suspensions are kept in a shaker at a fixed temperature of 30°C for 24 h, which is enough for guaranteeing adsorption equilibrium of MB over BL-ACs surface. The AC suspensions were next filtered, and the residual MB contained in the solution was quantified by UV–visible spectrophotometry (Unico™ UV-2000 spectrophotometer) working at a fixed wavelength of 662 nm [38], and using pre-plot standard curve.

The MB adsorption capacity at equilibrium, $Q_e$ (mg g$^{-1}$) was calculated using the following equation:

$$Q_e = \frac{(C_o - C_e)V}{W},$$

where $C_o$ and $C_e$ (mg l$^{-1}$) are the liquid-phase concentration of MB at initial and equilibrium, $V$ (l) is the volume of the MB solution, and $W$ (g) is the weight of the agro-based AC.

*Batch equilibrium studies.* The analysis of the isotherm data by fitting them to different isotherm models is an important step to find the suitable model that can be used for design purposes. Langmuir and Freundlich isotherms are the most common ones. The Langmuir theory is valid for monolayer adsorption onto a surface containing a finite number of identical sites. The linear form of Langmuir isotherm equation is expressed as [39]

$$\frac{C_e}{q_e} = \frac{1}{bq_m} + \frac{C_e}{q_m},$$

where $q_e$ is the amount adsorbed at equilibrium, $C_e$ is the equilibrium concentration of the adsorbate (MB), $q_m$ (mg g$^{-1}$) is the maximum adsorption capacity, and $b$ is the binding constant which is related to the heat of adsorption.

The Freundlich isotherm model is valid for heterogeneous surfaces. The linear form of Freundlich model is generally represented as follows [40]:

$$\log q_e = \log K_F + \frac{1}{n} \log C_e,$$

where $K_F$ and $n$ are Freundlich constants, $n$ giving an indication of how favourable the adsorption process and $K_F$ (mg g$^{-1}$ (l mg$^{-1}$)$^n$) is the adsorption capacity of the adsorbent.

The applicability of the isotherm equation is compared by judging the correlation coefficients $R^2$. The linear form of Langmuir's isotherm model, when $C_e/q_e$ is plotted against $C_e$, straight line with slope $1/Q_m$ is obtained, indicating that the adsorption of MB on BL-ACs follows the Langmuir isotherm. But for Freundlich isotherm model, by plotting $\log q_e$ against $\log C_e$ a straight line with slope $1/n$ is obtained. The Langmuir constants '$b$' and '$Q_m$' and Freundlich constant $K_F$ and $n$ are calculated from these isotherms and their values are collected. The essential characteristics of the Langmuir isotherm can be expressed in terms of a dimensionless equilibrium parameter ($R_L$), which is defined by

$$R_L = \frac{1}{1 + bC_m},$$

where $b$ is the Langmuir constant and $C_m$ is the highest dye concentration (mg l$^{-1}$). The value of $R_L$ indicates the type of the isotherm to be either unfavourable ($R_L > 1$), linear ($R_L = 1$), favourable ($0 < R_L < 1$) or irreversible ($R_L = 0$). The value of $R_L$ was ($0 < R_L < 1$) and confirmed that the BL-ACs of all samples is favourable for adsorption of MB dye under the conditions used in this study.

*Batch kinetic studies.* The kinetics of adsorption of MB on BL-ACs can be studied by applying the Lagergren first-order, pseudo-second-order and intraparticle diffusion models. The rate equations that have been most widely used for the adsorption of an adsorbate from an aqueous solution, which are expressed by the following equations [41–43]:

| kinetic model | linear form | plots | Ref. |
|---|---|---|---|
| Lagergren first-order | $\ln(q_e - q_t) = \ln q_e - K_1 t$ | $\ln(q_e - q_t)$ versus $t$ | [41] |
| pseudo-second-order | $\frac{t}{q_t} = \left[\frac{1}{K_2 q_e^2}\right] + \frac{1}{q_e} t$ | $t/q_e$ versus $t$ | [42] |
| intraparticle diffusion | $q_t = K_{id} t^{1/2} + C$ | $q_t$ versus $t^{1/2}$ | [43] |

where $q_e$ and $q_t$ are the amount of dye adsorbed per unit mass of the adsorbent (in $mg\,g^{-1}$) at equilibrium time and time $t$, respectively, $K$ is the rate constant, $C$ is the intraparticle diffusion constant.

# 3. Results and discussion

## 3.1. Black liquor characterization

### 3.1.1. Elemental analysis

The main elements (C, N, H, O and S) of the BLs which resulted from pulping the RS and SCB fibres are shown in table 1. These contents are related to chemical constituents of black liquors, which are mainly the delignified lignin, carbohydrate (mostly hydrolysed hemicelluloses) and sulfur of pulping reagents. The carbon content (% C) of the black liquor samples obtained from soda of bagasse (B-SP) is greater than the BL from neutral pulping (B-NSP) and the sulfite pulping lies in the later sequence (B-SP). But, for the case of BLs from RS, the sequence of carbon content is RS-NSP > RS-SP > RS-SP. The H/C and O/C ratios are used to measure the degree of aromaticity [44,45], as well as the degree of delignification and hydrolysis of hemicelluloses; therefore, increasing the lignin content together with hemicelluloses in BL will be accompanied by decreasing the H/C and O/C. This view is emphasized from comparing these ratios in the case of alkaline pulping of bagasse in comparison with alkaline pulping of RS. For the case of black liquors produced from sulfite and neutral sulfite of bagasse, the sulfur content affects this sequence in relation to delignification %. As the RS-BLs include different silica contents, as is clear from reducing the value of ash in pulp fibres compared to raw RS (table 1), the values of H/C and O/C in the case of alkaline pulping (RS-SP) are higher than those of BL (RS-SSP). The lower removal of silica by neutral sulfite pulping of RS provides BL with H/C and O/C that are nearly equivalent to B-NSP. Because $SO_3Na$ included lignin of BL-RS-NSP (with higher delignification), its % C is lower than that of soda pulping. From these foregoing data, we suggested that the BL with higher % C is convenient to provide ACs. To emphasize this opinion, further studies will be carried out (§3.2).

### 3.1.2. Thermal analysis (TGA/DTG)

Figure 2a–c and table 2 show the thermal analysis of the investigated BLs, as precursors of activated carbon. This study is carried out to indicate the trend of weight loss of BL versus carbonization temperature. The role of pulping processes on weight loss and consequently the char yield (weight remain) as a function of temperature are also examined in comparison with the studies previously performed on pulp fibres [35] and illustrated in figure 2a–c. TGA and DTG curves show that at lower temperature (less than 120°C), the weight loss is related to evaporation of moisture contents; while at relatively higher temperatures (in the range 117–376°C), most of the organic components of BLs are thermally decomposed (volatilized), and leaving the comparable amount of chars. With regard to DTG peak temperature of major degradation temperature, it is clear that BL from soda pulping of bagasse has higher peak temperature than DTG of B-BL-SSP and B-BL-NSP; while for BLs of RS pulping, the RS-BL-NSP has higher DTG peak temperature than other BLs. This is ascribed to the content of delignified lignin in BL. The appearance of two main degradation stages in black liquors from neutral sulfite pulping of both RS and SCB, in addition to absorbed water, is probably related to the degradation of low molecular weight hydrolysed carbohydrate followed by degradation of lig-$SO_3Na$ in BLs. The relatively higher sulfonyl content in the case of BL-RS-NSP is accompanied by increasing its DTG peak temperature (278.1°C) than the case of BL-B-NSP (232°C). The higher weight remains in RS-BLs (69–44%) than in B-BLs (48–18%), with strong difference in BLs from neutral sulfite pulping process. This observation is mainly ascribed to inorganic elements which included the BLs (sulfur and silica) [46,47].

On comparing the observed TGA of BLs with that observed of raw agro-substrates (RS and SCB) and their pulp fibres produced from the same pulping processes, figure 2a–c, it is clear that the weight loss of raw RS and SCB together with their pulps are greater than that of BLs. This is attributed to the higher organic matter contents, especially cellulose (C and H) of raw and pulped fibres as compared with BLs. Here, the degradation of cellulose leads to the formation of levoglucosan which promotes the volatilization stage. This difference is greater in the soda pulping of RS.

**Table 1.** Elemental analyses of black liquors of RS and SCB versus chemical analyses of pulps.

| code | black liquors | | | | | | pulped fibres | | | | | |
|---|---|---|---|---|---|---|---|---|---|---|---|---|
| | C % | S % | H % | H/C | O % | O/C | lignin % | cellulose % | hemicelluloses % | change in lignin, % | change in hemicellulose, % | Ash % |
| RS-cont. | — | — | — | — | — | — | 14.5 | 37.5 | 22.8 | — | — | 18.4 |
| RS-SP | 17.57 | — | 3.02 | 0.17 | 77.48 | 4.41 | 12.6 | 46.3 | 27.5 | −12.90 | 20.61404 | 14.8 |
| RS-SSP | 14.05 | 11.07 | 1.95 | 0.14 | 73.60 | 5.24 | 12.8 | 36.2 | 31.4 | −11.47 | 37.7193 | 15.6 |
| RS-NSP | 19.01 | 9.74 | 3.35 | 0.17 | 65.77 | 3.46 | 12.3 | 39.2 | 29.0 | −15.29 | 27.19298 | 17.5 |
| B-cont. | — | — | — | — | — | — | 19.1 | 41.6 | 26.5 | — | — | 4.7 |
| B-SP | 31.31 | — | 4.03 | 0.13 | 62.95 | 2.01 | 14.0 | 52.6 | 23.0 | −26.95 | −13.2075 | 1.3 |
| B-SSP | 17.68 | 8.71 | 2.62 | 0.15 | 70.03 | 3.96 | 18.1 | 44.0 | 26.9 | −5.26 | 1.509434 | 1.5 |
| B-NSP | 19.74 | 4.39 | 3.84 | 0.17 | 70.99 | 3.59 | 15.9 | 48.8 | 23.1 | −16.82 | −12.8302 | 1.8 |

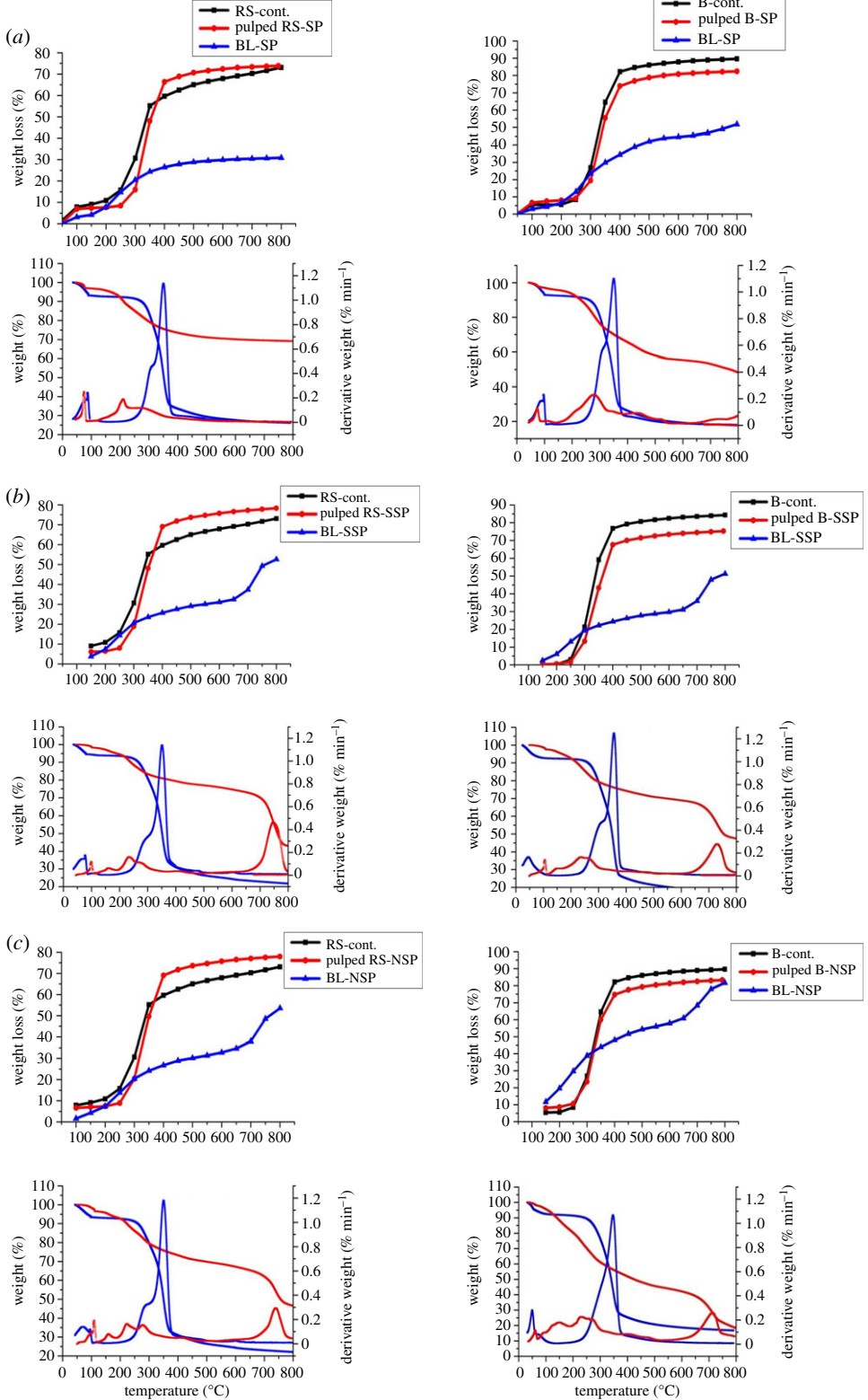

**Figure 2.** TGA/DTG analysis of BLs produced from (*a*) soda pulping, (*b*) sulfite pulping, and (*c*) neutral sulfite pulping of RS and SCB in comparison with produced pulp fibres.

## 3.2. Activated carbon characterization

### 3.2.1. Surface textural characterization

Figure 3 shows the adsorption isotherm and pore size distribution of the nitrogen gas at −196°C (77 K) of the BL-ACs and their textural characterization. These results are also given in table 3. As seen from figure 3, the

**Table 2.** DTG/TGA peak analysis of black liquors (BLs) of different pulped RS and SCB fibres.

| code | temperature range (°C) | DTG (°C) | weight remain (%) |
| --- | --- | --- | --- |
| BL-RS-SP | 44.8 – 87.9 | 74.1 | 96.9 |
| | 87.9 – 236.1 | 211.0 | 86.8 |
| BL-RS-SSP | 44.4 – 111.1 | 99.5 | 98.2 |
| | 189.2 – 365.6 | 234.6 | 80.4 |
| | 661.2 – 815.8 | 750.1 | 42.6 |
| BL-RS-NSP | 70.7 – 122.6 | 108.8 | 96.4 |
| | 122.6 – 184.8 | 159.4 | 93.4 |
| | 184.8 – 375.6 | 278.1 | 74.4 |
| | 638.6 – 796.6 | 741.5 | 46.6 |
| BL-B-SP | 42.6 – 95.2 | 73.7 | 97.1 |
| | 161.5 – 345.2 | 279.6 | 70.6 |
| BL-B-SSP | 48.7 – 117.4 | 105.8 | 97.5 |
| | 117.4 – 374.5 | 236.9 | 75.2 |
| | 621.1 – 815.9 | 731.3 | 46.9 |
| BL-B-NSP | 35.1 – 69.7 | 62.1 | 97.9 |
| | 69.7 – 197.8 | 150.2 | 88.3 |
| | 197.8 – 313.6 | 231.9 | 59.6 |
| | 613.1 – 793.3 | 715.8 | 18.7 |

adsorption isotherm of the BL-AC is greatly affected by the type of pulping process. But the isotherms obtained from ACs using soda pulping black liquors as precursors (RS-BL-SP and B-BL-SP) follow the adsorption isotherms of type I, according to the IUPAC classification. The nitrogen uptake by ACs slightly increased at very low relative pressure, followed by slightly adsorption increment in a plateau form with an increase in relative pressure to 1, indicating the existence of microporosity. This view is emphasized from the values of $V_{mic}/V_T$ (table 3). But, the isotherms of ACs from BLs of sulfite and neutral sulfite pulping of both RS and SCB (BL-SSP and BL-NSP) provide adsorption isotherms of types I and II, which are reflected in the presence of some mesopores in the carbon structure [48].

It can be noted that AC with relatively higher surface area can be produced from the black liquors of neutral sulfite pulping process of both RS and SCB (927–921 $m^2\,g^{-1}$). However, a lower surface area is noted for sulfite pulping of RS (BL-RS-SSP; 215 $m^2\,g^{-1}$). It is interesting to note that (figure 4 and table 3) the specific surface area ($S_{BET}$) and total pore volume ($V_T$) of the ACs produced from RS-BL-NSP are greater than that produced from its pulp fibres. The reverse trend is noted in BLs from SCB as well as soda and sulfite pulping RS, where the $S_{BET}$ of ACs prepared from RS pulps of neutral sulfite pulping is 440 $m^2$ g; while that produced from BL is 921 $m^2\,g^{-1}$. Also, the $V_T$ of neutral sulfite pulp-AC and its BL-AC are 0.79 and 025 $cm^3\,g^{-1}$, respectively. The opposite trend is observed in ACs from SCB pulp fibres versus their black liquors (figure 4). The lowest $S_{BET}$ of RS-BL-SSP-AC may start decreasing in % C content which resulted from lowering the delignification, as compared with other pulping cases. The BL-ACs of the neutral pulping RS and SCB possess the maximum surface area (928 and 921 $m^2\,g^{-1}$), with a total pore volume of 0.701 and 0.79 $cm^3\,g^{-1}$, respectively.

Based on the textural characterization of the ACs produced from black liquors, it can be recommended that the by-products of neutral sulfite pulping are the best ones for producing high efficient BL-ACs. As can be seen that the behaviour of these investigated RS-BL-NSP and B-BL-NSP are nearly similar to those produced from the literature reported lignin, using physical and chemical activation approaches [14,49–51].

### 3.2.2. FTIR spectra and SEM

The changes in surface function groups and morphology of the investigated BL-based ACs according to the pulping process are illustrated in figures 6 and 7. With regard to FT-IR spectra (figure 5a,b), it is clear

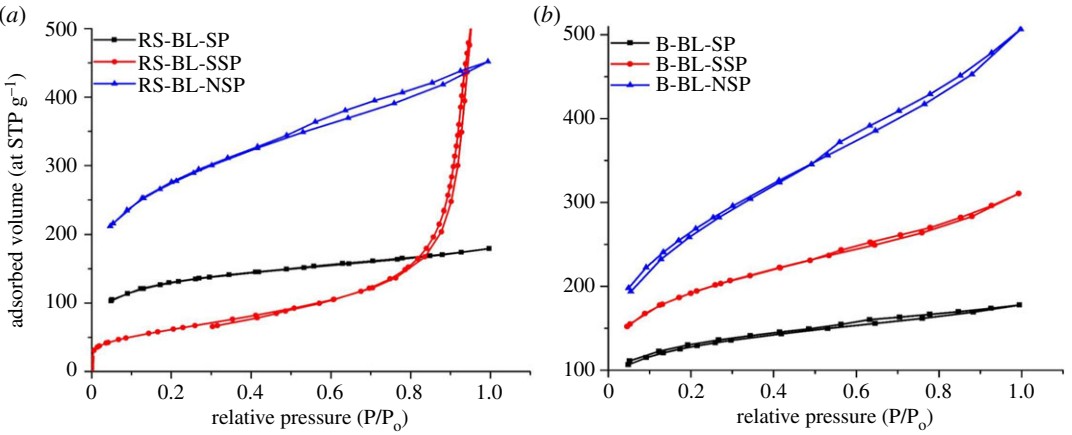

**Figure 3.** Adsorption isotherms of nitrogen on various BL-ACs.

**Table 3.** Textural characterization of various BL-ACs.

| code | $S_{BET}$ (m$^2$ g$^{-1}$) | $S_{mic}$ (m$^2$ g$^{-1}$) | $S_{BJH}$ (m$^2$ g$^{-1}$) | $V_{BJH}$ (cm$^3$ g$^{-1}$) | $V_{T\,(0.95)}$ (cm$^3$ g$^{-1}$) | $V_{mic}$ (cm$^3$ g$^{-1}$) | $V_{mic}/V_T$ | average pore radius (nm) |
|---|---|---|---|---|---|---|---|---|
| BL-RS-SP | 425.77 | 360.99 | 40.09 | 0.073 | 0.278 | 0.173 | 62.23 | 1.31 |
| BL-RS-SSP | 214.99 | 143.3 | 183.52 | 0.738 | 0.794 | 0.033 | 4.16 | 7.38 |
| BL-RS-NSP | 927.66 | 704.33 | 148.49 | 0.272 | 0.701 | 0.337 | 48.07 | 1.51 |
| BL-B-SP | 415.49 | 351.4 | 41.14 | 0.074 | 0.275 | 0.172 | 62.55 | 1.33 |
| BL-B-SSP | 641.07 | 493.44 | 92.84 | 0.185 | 0.482 | 0.237 | 49.17 | 1.5 |
| BL-B-NSP | 920.78 | 633.42 | 198.1 | 0.381 | 0.786 | 0.299 | 38.04 | 1.71 |

that the surface of BL-ACs is characterized with a strong band at about 3468 cm$^{-1}$ assigned to O-H stretching of phenolic lignin, carboxylic acid or adsorbed water [52]. The low intensity peak at 2940 cm$^{-1}$ was assigned to CH asymmetric stretching. There are sharp-weak bands at 2398 cm$^{-1}$ assigned to C=N. The peaks at 1640 and 1210 cm$^{-1}$ were assigned to C=O stretching and ether group C–O–C linkage, respectively. The sharp band of the ether observed in BL-RS-SP is related to the lignin, hemicellulose or due to the stretching of the Si–O–Si bonds [53], because soda pulping leads to higher removing of silica from RS, and consequently higher silica content in BL. The bands at 525 and 720 cm$^{-1}$ were assigned to C=C stretching and bending of benzene ring. Due to higher lignin including B-BLs (higher delignification), with lower silica content, the band peaks at 1607–1624 cm$^{-1}$ (vibration of aromatic hydrocarbons [54]) are more intense than in RS-BL-ACs. This indicates that the surface of activated carbon characterized with different functional groups is ready for the adsorption process.

For the case of SEM study, the morphology of BL-ACs from RS shows inhomogeneous and rough surface (figure 6). At the magnification 40 000, it can be seen that the surface is covered by silicophosphate crystals. The AC from soda pulping black liquor of RS has higher silicophosphate crystals; however, lower crystal is noted in AC from BL of neutral sulfite pulping of RS. This observation is related to the reduction of ash content of RS as a result of pulping processes (table 1). Owing to the highest inorganic crystals of soda BL-AC its surface area is lower than that of NSP (table 3). It can be clearly seen from figure 6 that the distribution of micropores in graphs of ACs produced from SCB-BLs is greatly affected by the approach of pulping and emphasized the data of textural characterization reported in table 3, i.e. SCB-AC with higher surface area shows highly porous in morphology image (BL-B-NSP).

### 3.2.3. Iodine value

Figure 7a shows the adsorption of iodine on the surface of ACs, which are produced from black liquors of RS and SCB pulping processes. These values are also compared with those produced from previously synthesized pulp fibres [35]. It can be seen that BL-based AC from neutral sulfite pulping of RS shows

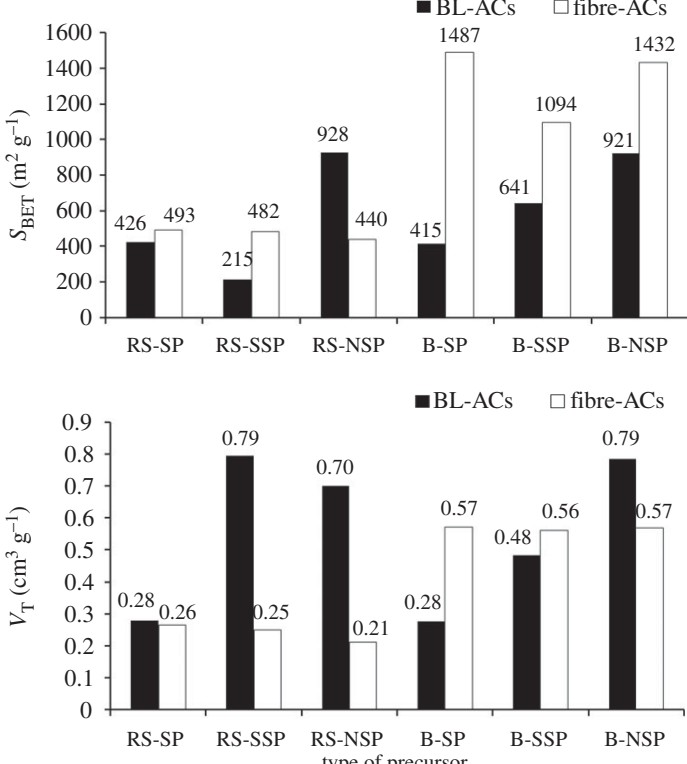

**Figure 4.** The specific surface area and total pore volume of ACs prepared from the pulped RS and SCB fibres and their black liquors.

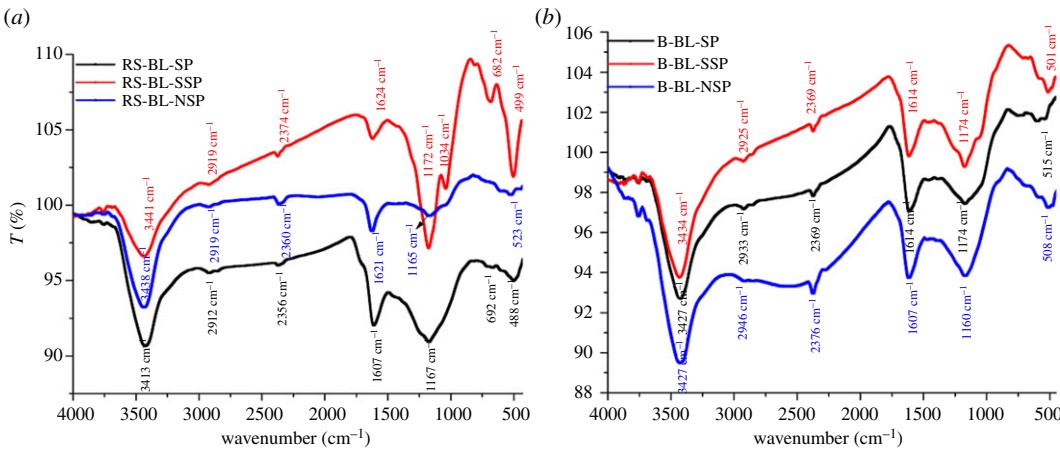

**Figure 5.** (*a*) FTIR analysis of RS-BL-ACs. (*b*) FTIR analysis of B-BL-ACs.

higher efficiency in removing the iodine (928 mg g$^{-1}$), in comparison with BL-ACs of other pulping types and those produced from pulps. This is related to its superiority of specific surface area $S_{BET}$ and total pore volumes. However, the opposite trend is observed with the SCB pulp-ACs versus their black liquors. Here, the ACs produced from SCB pulps of soda, sulfite and neutral sulfite pulping have I$_2$-values 1402, 1243 and 1384 mg g$^{-1}$, respectively; while its values are 899, 901 and 928 mg g$^{-1}$ in ACs produced from their BLs. As can be seen, a direct proportional relationship existed between the adsorption of iodine (i.e. iodine number) and the specific surface area $S_{BET}$ of RS- and SCB-black liquor-ACs. The foregoing data of I$_2$-values support the previous suggestion; the neutral sulfite pulping is featured for producing highly efficient BL-AC.

### 3.2.4. Batch equilibrium and kinetic studies of MB adsorption

The equilibrium and kinetic adsorption of MB dye on the BL-ACs of RS and SCB are studied, and the finding results are compared with previous data dealing with the ACs from RS and SCB pulps [35]

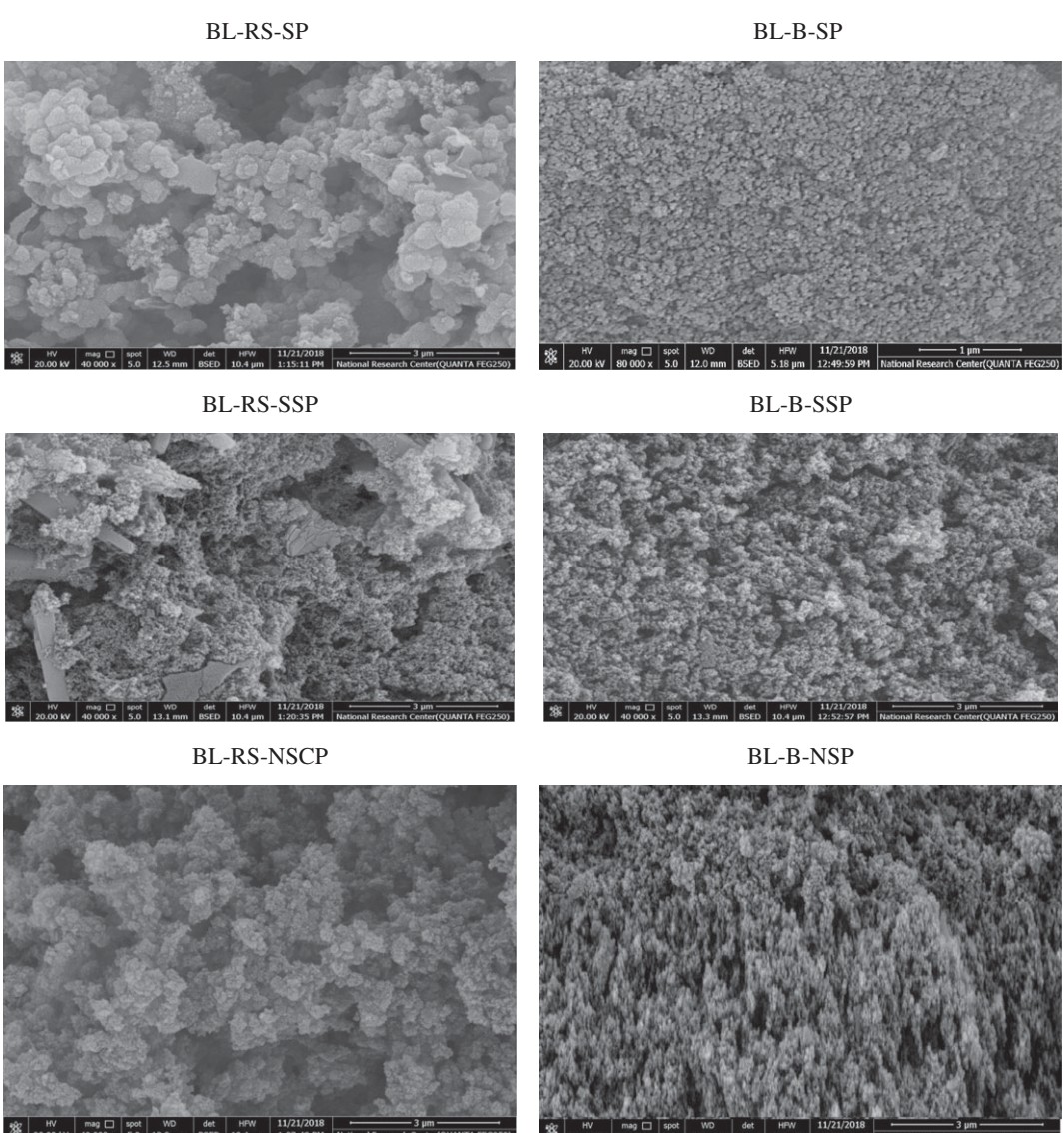

**Figure 6.** SEM of BL-RS-ACs and BL-B-ACs.

**Table 4.** Langmuir and Freundlich isotherm parameters for adsorption of MB dye onto ACs prepared from black liquor of different pulped RS and SCB fibres.

| | Langmuir isotherm | | | | | | Freundlich isotherm | | | |
|---|---|---|---|---|---|---|---|---|---|---|
| | slope | $Q_m$ | Inter. | $b$ | $R^2$ | $R_L$ | $1/n$ = slope | Inter. | $K_F$ | $R^2$ |
| BL-RS-SP | 0.0037 | 270.3 | 0.0008 | 4.63 | 1 | 0.00036 | 0.555 | 2.43 | 269.77 | 0.853 |
| BL-RS-SSP | 0.00956 | 104.6 | 0.0096 | 1.00 | 0.99 | 0.00166 | 0.0973 | 1.76 | 56.89 | 0.981 |
| BL-RS-NSP | 0.0027 | 370.4 | 0.0017 | 1.63 | 0.963 | 0.00102 | 0.577 | 2.35 | 222.13 | 0.819 |
| BL-B-SP | 0.00408 | 245.1 | 0.0008 | 5.16 | 0.999 | 0.00032 | 0.1616 | 2.15 | 141.35 | 0.576 |
| BL-B-SSP | 0.00423 | 236.4 | 0.0029 | 1.46 | 0.998 | 0.00114 | 0.1427 | 2.12 | 132.74 | 0.583 |
| BL-B-NSP | 0.0042 | 238.1 | 0.0035 | 1.19 | 0.999 | 0.00140 | 0.1728 | 2.08 | 120.14 | 0.761 |

(tables 4 and 5 and figure 7$b$). Table 4 shows the experimental data of the equilibrium adsorption using Langmuir and Freundlich models. The adsorption can be described by Langmuir model with approximately correlation coefficient values ($R^2$) greater than 0.99. In addition to the values of $R^2$, the dimensionless equilibrium parameter ($R_L$) is found between 0 and 1, which indicates the favourable adsorption process. The values of ($1/n$) of Freundlich model indicate also favour the adsorption

**Table 5.** Lagergren first-order, pseudo-second-order kinetic model and intraparticle diffusion parameters for adsorption of MB onto ACs prepared from different raw fibres without pulping and pulped RS and SCB fibres.

| code | $Q_e$ (exp) | lagergren first-order model | | | | | | pseudo-second-order | | | | | | intraparticle diffusion | | | |
|---|---|---|---|---|---|---|---|---|---|---|---|---|---|---|---|---|---|
| | | $K_1$ ($h^{-1}$) | I | $q_{eq}$ (mg $g^{-1}$) | $R^2$ | SEE | I | $K_2$ ($h^{-1}$) | S | $q_{eq}$ (mg $g^{-1}$) | $R^2$ | SEE | $K_{id}$ | C | $R^2$ | SEE |
| BL-RS-SP | 249.4 | 0.289 | 3.49 | 32.98 | 0.996 | 0.119 | 0.0004 | 0.040 | 0.0040 | 250.00 | 1.000 | 0.0002 | 5.48 | 225.41 | 0.82 | 4.03 |
| BL-RS-SSP | 121.7 | 0.178 | 3.16 | 23.57 | 0.920 | 0.368 | 0.0031 | 0.021 | 0.0081 | 123.46 | 0.999 | 0.0009 | 5.806 | 96.039 | 0.79 | 4.75 |
| BL-RS-NSP | 249.9 | 0.267 | 3.29 | 26.84 | 0.980 | 0.245 | 0.0002 | 0.080 | 0.0040 | 250.00 | 1.000 | 0.0002 | 4.49 | 229.27 | 0.91 | 2.32 |
| BL-B-SP | 242.9 | 0.205 | 4.23 | 69.20 | 0.991 | 0.089 | 0.0007 | 0.024 | 0.0041 | 243.90 | 0.999 | 0.0007 | 13.22 | 182.04 | 0.96 | 4.09 |
| BL-B-SSP | 219.4 | 0.173 | 3.50 | 33.18 | 0.938 | 0.312 | 0.0011 | 0.019 | 0.0046 | 217.39 | 0.999 | 0.0002 | 8.71 | 181.56 | 0.67 | 9.82 |
| BL-B-NSP | 224.9 | 0.185 | 4.11 | 60.64 | 0.995 | 0.089 | 0.0012 | 0.016 | 0.0044 | 227.27 | 1.000 | 0.0006 | 12.85 | 166.51 | 0.91 | 6.12 |

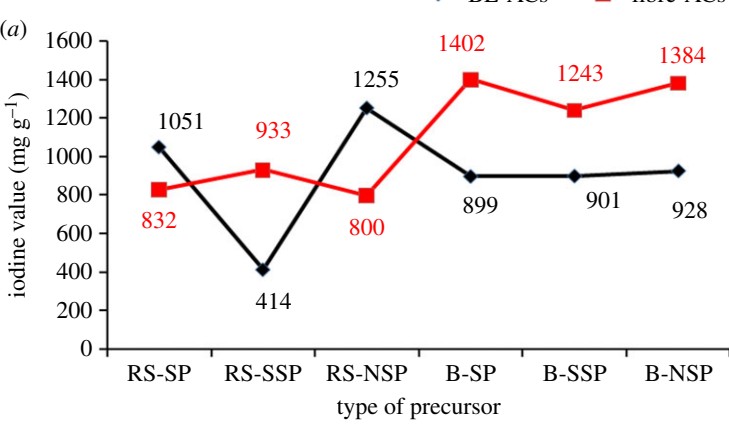

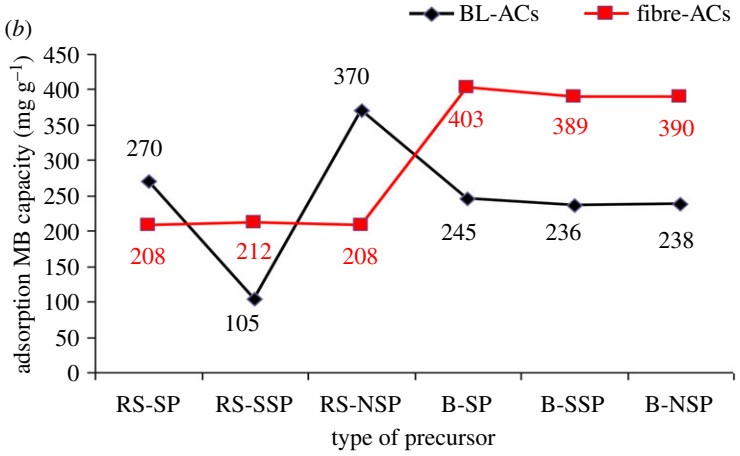

**Figure 7.** Adsorption capacity of ACs prepared from the pulped RS and SCB fibres and their black liquors toward (a) iodine and (b) MB.

**Table 6.** Comparing the adsorption behaviour of our present AC with literature ACs from isolated lignins.

| source of active carbon | activation or carbonization condition | $Q_{MB}$ (mg g$^{-1}$) | iodine number (mg g$^{-1}$) | Ref. |
|---|---|---|---|---|
| black liquor lignin as a waste of the hydrolysis wood processing | activated at 800°C for 60 min | 294 | 954 | [55] |
| black liquor lignin obtained from Bilt Graphic Paper Products | activated using H$_3$PO$_4$ at 800°C for 60 min | 166 | 1300 | [56] |
| black liquor lignin obtained from wood chips | activated at 900°C for 90 min | 524 | —— | [57] |
| organosolv lignin obtained from steam-exploded RS | no activation | 40 | —— | [58] |
| black liquor obtained from black liquor of neutral sulfite pulping of RS | activated using H$_3$PO$_4$ at 450°C for 60 min | 370 | 1255 | present work |

process of MB; however, their $R^2$ is lower than that of Langmuir model. So the MB adsorption is supported to be described by Langmuir model. According to this model, the maximum MB adsorption is observed in ACs from BL-RS-NSP (370 mg g$^{-1}$) and BL-B-NSP (245 mg g$^{-1}$). The efficiency of BLs-based ACs toward MB adsorption in comparison with pulp fibres-based ACs, behaves the same as the case of I$_2$-values (figure 5b). The ACs from BLs of soda and neutral sulfite

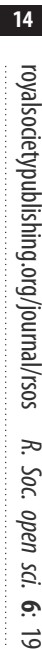

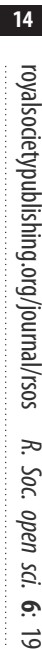

**Figure 8.** Kinetics and isotherm of BL-AC obtained from neutral sulfite of RS.

pulping of RS have greater affinity to adsorb MB than the ACs produced from their pulps; however, the reverse is observed in SCB-ACs.

The benefit of using the black liquor from neutral sulfite pulping as precursor for production of AC is proved when its adsorption capacity of MB and $I_2$ are compared with the previous published articles dealing with lignin-based ACs (table 6). Here, the adsorption capacity of BL-RS-AC surpasses the ACs investigated in most literature that reported on lignin-based ACs [57–60].

The kinetics of MB adsorption onto the investigated ACs are studied by applying three models, i.e. Lagergren first-order, pseudo-second-order and intraparticle diffusion. Figure 8 represents the kinetics and isotherms of BL-RS-AC, which provided greatest adsorption, as an example. From table 5, it is clear that the values of correlation coefficient $R^2$ of the pseudo-second-order model (0.999–1) are higher than those calculated from the other two models. Also the computed adsorption capacity values ($q_e$) according to pseudo-second-order model are in agreement with experimental values ($Q_e$(exp)). Finally, the standard error of estimate values (SEE) of the pseudo-second-order model are lower than the first-order and intraparticle diffusion models. From the foregoing results, it is expected that the adsorption of the MB on the surface of investigated RS-BL-ACs and SCB-BL-ACs follows the pseudo-second-order mechanism.

## 4. Conclusion

In the present study, a new approach was carried out to enhance the use of black liquors produced from the pulping of RS and SCB. Using these by-products as precursor for the production of ACs will preserve

the environment from disposing such effluents. Electiveness of agro-pulping process was estimated from pulping the RS by three pulping processes; namely, soda, sulfite and neutral sulfite, and assessing the produced BLs by-products in the production of ACs. The changes in pulping process accompanied the change in degree of delignification and hemicellulose in RS and SCB fibres; accordingly, it has a profound effect on BL constituents, and porous structure of produced ACs, using $H_3PO_4$ acid as chemical activating agent. This study was useful in providing ACs with surface area 920.8 and 927.7 $m^2 g^{-1}$, in RS and SCB, respectively. The maximum iodine number and MB adsorption capacity (1255 and 370 $mg g^{-1}$) were observed in ACs from BL of neutral sulfite pulping of RS. Moreover, it is interesting to note that the BL of neutral sulfite pulping of RS produced efficient AC toward both $S_{BET}$ and adsorption capacities than that produced from its pulp fibres. It is interesting to note that the AC provided from RS-NSP has greater adsorption capacity for $I_2$ and MB than that produced from RS-pulp fibres.

Ethics. This work was carried out by corresponding author and co-author, in continuation to their research activities.
Data accessibility. All data are contained in the article.
Authors' contributions. The corresponding author and co-author shared in idea, practical and writing of this research work.
Competing interests. We declare we have no competing interests.
Funding. Funding is self-employed from all authors.

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
