## [Reviewer comments · Royal Society Open Science]

Review History

RSOS-190173.R0 (Original submission)

Review form: Reviewer 1

Is the manuscript scientifically sound in its present form?

No

Are the interpretations and conclusions justified by the results?

Yes

Is the language acceptable?

No

Is it clear how to access all supporting data?

Yes

Do you have any ethical concerns with this paper?

No

Have you any concerns about statistical analyses in this paper?

No

Recommendation?

Major revision is needed (please make suggestions in comments)

Comments to the Author(s)

Revision for Royal Society Open Science

Manuscript: RSOS-190173 – Electiveness of agro-pulping process in the sustainable production of black liquor-based activated carbons

Corresponding Author: AH Basta

All authors: VF Lotfy and AH Basta

Present manuscript refers to the production of activated carbons using by-products. I think that the scope of this manuscript is important and valuable for the scientific community. However, some important aspects are missing and should be addressed in order to publish the present paper. Overall, English should be very carefully revised since it is difficult to point out all the weaknesses; I suggest the revision of the manuscript by a person whose mother language is English. Also, the aim and the novelty of the work should be clearer. My recommendation for this manuscript is major revision, based on the stated above and on the following:

Abstract

- First sentence: I suggest to changing it to “During the production of paper pulp, the waste water loaded with organic materials from pulping process is discharged.”
- Sentence “The trial process to utilize such effluent for production of activated carbon will be effective for omitting the wastewater treatment, together with obtaining value and demand product.” is confusing. Should be rephrased for clarity.
- Refer to SBET, as “specific surface area”.
- All the abbreviations should be defined, even those that are triter as MB, FTIR, SEM or ACs.
- Define NSP.
- Use the abbreviations SCB and RS for the first time when they are written in full (line 17).
- Avoid the use of “&”.
- Last sentence (“It interesting to note that BL disposed from RS-NSP provides AC is more efficient adsorption capacity for I2 and MB than that produced from RS- pulp fibers.”): correct the EN language.

Introduction

- Page 2, Line 46: Change “in production valuable products” to “the production of valuable products”.
- Page 2, Line 50: Change “agro-wastes paper pulping” to “paper pulping agro-wastes”.
- Page 2, Line 55: Define RS.
- Page 3, Line 8: Use “for synthesis of chemicals” or for synthesizing chemicals”.
- Page 3, Line 15 vs Line 19: Define activated carbons the first time they are mentioned (in line 15, not 19!). This issue with the abbreviations should be carefully revised in the whole manuscript!
- Page 3, Line 19: This sentence would be worthy more references.
- Page 3, Lines 34 and 38: Avoid the use of words as “my” and “we”.
- Page 3, Lines 36: Correct the typo.
- End of the introduction: The aim of the work should be clearer! Nowadays, there are plenty of works dealing with the production of ACs, using agro- or industrial-wastes. Which is the main driving force of this study? What is the novelty?
- The importance of the adsorbate chosen should be addressed. Why MB is important to be adsorbed? Why MB constitute a problem?

- Did the authors considered the circular economy approach? It would be interesting to link this type of work to that approach, since it deals with wastes and giving them an added-value.

Experimental

- Figure should be numbered and mentioned in the text.
- Page 4, Line 48: Avoid the term "our".
- Page 4, Line 50: BLs were already defined. Once again, the issue of the abbreviations should be carefully revised in the whole manuscript!
- Page 6, Line 23: Which are the reasons for that high concentration levels?
- Page 6, Line 29: How the authors performed the UV-vis measures? Was the matrix of the AC taken into account? The measures should be performed against the matrix. Please comment.
- Page 7, Line 49: Do the authors think that the kinetic models used may help in gathering knowledge about mechanisms of adsorption?
- Page 8, Line 19: Do the authors believe that the FTIR-KBr is the best option? Would not be better to use ATR-FTIR? Please comment.
- Page 8, Line 29: Correct the title of section 2.3.2.4 to "microscopy".

Results

- I wonder why the elemental analysis and the TGs were performed solely for the BLs and not for the ACs. I think it would be useful to do such characterization for the ACs as well. Please comment.
- Do the authors have the possibility of measure the TOC of the materials? It would be an important measure to complement the characterization.

Tables and Figures

- Figure 3 is a duplication of data from Table 3. Use either the Figure or the Table.
- Kinetics and isotherms should be presented in a figure.

Manuscript General Comment: I have to stress that the manuscript should be carefully revised, mostly to improve the English, but also to correct the countless typos (I started to address those but it is impossible for the referee to do such action when there are so many!). Also, attention should be paid to the abbreviations' issue that I've already stated but that happens countless times.

Review form: Reviewer 2 (Weigang Zhao)

Is the manuscript scientifically sound in its present form?

Yes

Are the interpretations and conclusions justified by the results?

Yes

Is the language acceptable?

Yes

Is it clear how to access all supporting data?

Not Applicable

Do you have any ethical concerns with this paper?

No

Have you any concerns about statistical analyses in this paper?

No

Recommendation?

Accept with minor revision (please list in comments)

Comments to the Author(s)

This manuscript presents a simple procedure to prepare cost-effective porous activated carbons from black liquor during production of paper pulp using phosphoric acid as activation agent. The result carbons were used as aqueous adsorbates. The work is interesting and the manuscript is well organized. However, the authors should address the questions below:

1. As the author said, there are lot of micropores for the ACs, so for the N₂ adsorption-desorption isotherms, more points at the low pressure range (<0,1) should be present.
2. For confirming, the pore size distribution from N₂ adsorption-desorption isotherms should also add in the text.
3. The structure of this paper should well organize. For exempla, the IR and SEM results should be in front of the adsorption results of iodine and MB.
4. There are lot of resent works about ACs for iodine and MB adsorption, so some new reference should add.
5. A new part about the comparison should also present.
6. The Conclusions likes an experiment summary rather a Conclusion which should give not only the main conclusion but also echoing the introduction.
7. The quality of "Scheme of the Experimental work" is too low. It happened also for all the equations in this paper.

Review form: Reviewer 3 (Packiyam Ravichandran)

Is the manuscript scientifically sound in its present form?

Yes

Are the interpretations and conclusions justified by the results?

Yes

Is the language acceptable?

Yes

Is it clear how to access all supporting data?

Not Applicable

Do you have any ethical concerns with this paper?

No

Have you any concerns about statistical analyses in this paper?

No

Recommendation?

Accept as is

Comments to the Author(s)

Dear Authors

Wonderful work have been done by you, and check any residue found after the Activated carbon production by is the black liquor.

Decision letter (RSOS-190173.R0)

01-Mar-2019

Dear Miss Basta:

Title: Electiveness of agro-pulping process in the sustainable production of black liquor-based activated carbons

Manuscript ID: RSOS-190173

The editor assigned to your manuscript has now received comments from reviewers. We would like you to revise your paper in accordance with the referee and Subject Editor suggestions which can be found below (not including confidential reports to the Editor). Please note this decision does not guarantee eventual acceptance.

Please submit your revised paper before 24-Mar-2019. Please note that the revision deadline will expire at 00.00am on this date. If we do not hear from you within this time then it will be assumed that the paper has been withdrawn. In exceptional circumstances, extensions may be possible if agreed with the Editorial Office in advance. We do not allow multiple rounds of revision so we urge you to make every effort to fully address all of the comments at this stage. If deemed necessary by the Editors, your manuscript will be sent back to one or more of the original reviewers for assessment. If the original reviewers are not available we may invite new reviewers.

Please also include the following statements alongside the other end statements. As we cannot publish your manuscript without these end statements included, if you feel that a given heading is not relevant to your paper, please nevertheless include the heading and explicitly state that it is not relevant to your work.

- Acknowledgements

RSC Associate Editor:
Comments to the Author:
(There are no comments.)

RSC Subject Editor:
Comments to the Author:
(There are no comments.)

Reviewers' Comments to Author:
Reviewer: 1

Comments to the Author(s)
Revision for Royal Society Open Science

Manuscript: RSOS-190173 – Electiveness of agro-pulping process in the sustainable production of black liquor-based activated carbons

Corresponding Author: AH Basta
All authors: VF Lotfy and AH Basta

Present manuscript refers to the production of activated carbons using by-products. I think that the scope of this manuscript is important and valuable for the scientific community. However, some important aspects are missing and should be addressed in order to publish the present paper. Overall, English should be very carefully revised since it is difficult to point out all the weaknesses; I suggest the revision of the manuscript by a person whose mother language is English. Also, the aim and the novelty of the work should be clearer. My recommendation for this manuscript is major revision, based on the stated above and on the following:

Abstract

- First sentence: I suggest to changing it to “During the production of paper pulp, the waste water loaded with organic materials from pulping process is discharged.”
- Sentence “The trial process to utilize such effluent for production of activated carbon will be

effective for omitting the wastewater treatment, together with obtaining value and demand product.” is confusing. Should be rephrased for clarity.

- Refer to SBET, as “specific surface area”.
- All the abbreviations should be defined, even those that are triter as MB, FTIR, SEM or ACs.
- Define NSP.
- Use the abbreviations SCB and RS for the first time when they are written in full (line 17).
- Avoid the use of “&”.
- Last sentence (“It interesting to note that BL disposed from RS-NSP provides AC is more efficient adsorption capacity for I2 and MB than that produced from RS- pulp fibers.”): correct the EN language.

Introduction

- Page 2, Line 46: Change “in production valuable products” to “the production of valuable products”.
- Page 2, Line 50: Change “agro-wastes paper pulping” to “paper pulping agro-wastes”.
- Page 2, Line 55: Define RS.
- Page 3, Line 8: Use “for synthesis of chemicals” or for synthesizing chemicals”.
- Page 3, Line 15 vs Line 19: Define activated carbons the first time they are mentioned (in line 15, not 19!). This issue with the abbreviations should be carefully revised in the whole manuscript!
- Page 3, Line 19: This sentence would be worthy more references.
- Page 3, Lines 34 and 38: Avoid the use of words as “my” and “we”.
- Page 3, Lines 36: Correct the typo.
- End of the introduction: The aim of the work should be clearer! Nowadays, there are plenty of works dealing with the production of ACs, using agro- or industrial-wastes. Which is the main driving force of this study? What is the novelty?
- The importance of the adsorbate chosen should be addressed. Why MB is important to be adsorbed? Why MB constitute a problem?
- Did the authors considered the circular economy approach? It would be interesting to link this type of work to that approach, since it deals with wastes and giving them an added-value.

Experimental

- Figure should be numbered and mentioned in the text.
- Page 4, Line 48: Avoid the term “our”.
- Page 4, Line 50: BLs were already defined. Once again, the issue of the abbreviations should be carefully revised in the whole manuscript!
- Page 6, Line 23: Which are the reasons for that high concentration levels?
- Page 6, Line 29: How the authors performed the UV-vis measures? Was the matrix of the AC taken into account? The measures should be performed against the matrix. Please comment.
- Page 7, Line 49: Do the authors think that the kinetic models used may help in gathering knowledge about mechanisms of adsorption?
- Page 8, Line 19: Do the authors believe that the FTIR-KBr is the best option? Would not be better to use ATR-FTIR? Please comment.
- Page 8, Line 29: Correct the title of section 2.3.2.4 to “microscopy”.

Results

- I wonder why the elemental analysis and the TGs were performed solely for the BLs and not for the ACs. I think it would be useful to do such characterization for the ACs as well. Please comment.
- Do the authors have the possibility of measure the TOC of the materials? It would be an important measure to complement the characterization.

Tables and Figures

- Figure 3 is a duplication of data from Table 3. Use either the Figure or the Table.

- Kinetics and isotherms should be presented in a figure.

Manuscript General Comment: I have to stress that the manuscript should be carefully revised, mostly to improve the English, but also to correct the countless typos (I started to address those but it is impossible for the referee to do such action when there are so many!). Also, attention should be paid to the abbreviations' issue that I've already stated but that happens countless times.

Reviewer: 2

Comments to the Author(s)

This manuscript presents a simple procedure to prepare cost-effective porous activated carbons from black liquor during production of paper pulp using phosphoric acid as activation agent. The result carbons were used as aqueous adsorbates. The work is interesting and the manuscript is well organized. However, the authors should address the questions below:

1. As the author said, there are lot of micropores for the ACs, so for the N₂ adsorption-desorption isotherms, more points at the low pressure range (<0,1) should be present.
2. For confirming, the pore size distribution from N₂ adsorption-desorption isotherms should also add in the text.
3. The structure of this paper should well organize. For exempla, the IR and SEM results should be in front of the adsorption results of iodine and MB.
4. There are lot of resent works about ACs for iodine and MB adsorption, so some new reference should add.
5. A new part about the comparison should also present.
6. The Conclusions likes an experiment summary rather a Conclusion which should give not only the main conclusion but also echoing the introduction.
7. The quality of "Scheme of the Experimental work" is too low. It happened also for all the equations in this paper.

Reviewer: 3

Comments to the Author(s)

Dear Authors

Wonderful work have been done by you, and check any residue found after the Activated carbon production by is the black liquor.

Author's Response to Decision Letter for (RSOS-190173.R0)

See Appendix A.

Decision letter (RSOS-190173.R1)

26-Mar-2019

Dear Miss Basta:

Title: Electiveness of agro-pulping process in the sustainable production of black liquor-based activated carbons

Manuscript ID: RSOS-190173.R1

It is a pleasure to accept your manuscript in its current form for publication in Royal Society Open Science. The chemistry content of Royal Society Open Science is published in collaboration with the Royal Society of Chemistry.

RSC Associate Editor
Comments to the Author:
(There are no comments.)

Reviewer(s)' Comments to Author:

Appendix A

Dear Editor Prof. Anthony Stace and Associate Editor Dr Ya-Wen Wang

Thank you for your e.mail and the reviewers' comments helping us to improve our manuscript concerning our manuscript with title "Electiveness of agro-pulping process in the sustainable production of black liquor-based activated carbons" (Manuscript ID: RSOS-190173). All of the comments have been discussed and illustrated in the revised article, highlighted by grey color. Please, see bellow the list of answers to comments, and we hope (LORD willing) that the correction will meet with approval.

Reviewers' comments and our answers:

Reviewer #1:

1. Comments related to Abstract section

- First sentence: I suggest to changing it to "During the production of paper pulp, the waste water loaded with organic materials from pulping process is discharged."
- Sentence "The trial process to utilize such effluent for production of activated carbon will be effective for omitting the wastewater treatment, together with obtaining value and demand product." is confusing. Should be rephrased for clarity.
- Refer to SBET, as "specific surface area".
- All the abbreviations should be defined, even those that are triter as MB, FTIR, SEM or ACs.
- Define NSP.
- Use the abbreviations SCB and RS for the first time when they are written in full (line 17).
- Avoid the use of "&".
- Last sentence ("It interesting to note that BL disposed from RS-NSP provides AC is more efficient adsorption capacity for I2 and MB than that produced from RS- pulp fibers."): correct the EN language.

Answer

We modified and added the full names together with abbreviations in the revised version of Ms, as reviewer asked. All changes are highlighted, with thanks.

2. Comments related to Introduction section

- Page 2, Line 46: Change "in production valuable products" to "the production of valuable products".
- Page 2, Line 50: Change "agro-wastes paper pulping" to "paper pulping agro-wastes".
- Page 2, Line 55: Define RS.
- Page 3, Line 8: Use "for synthesis of chemicals" or for synthesizing chemicals".
- Page 3, Line 15 vs Line 19: Define activated carbons the first time they are mentioned (in line 15, not 19!). This issue with the abbreviations should be carefully revised in the whole manuscript!
- Page 3, Line 19: This sentence would be worthy more references.

Answer:

I made the forgoing changes. With regard to his (her) last comment we added 3 relevant references for articles published FY 2013, 2018, and 2019, with thanks. The References, became [19-23] instead of [19, 20]. All further References are shifted

- Page 3, Lines 34 and 38: Avoid the use of words as "my" and "we".
- Page 3, Lines 36: Correct the typo.

Answer,

The corrections are made as the reviewer asked

- End of the introduction: The aim of the work should be clearer! Nowadays, there are plenty of works dealing with the production of ACs, using agro- or industrial-wastes. Which is the main driving force of this study? What is the novelty?

Answer

We replaced the last paragraph of Introduction section by the following sentences.

In this present article the black liquors as a whole, which resulted from pulping of rice straw are used for production of ACs. The type of pulping processes of RS (alkaline, acidic and neutral) are optimized, and compared with BLs produced from sugar-cane bagasse (SCB). Moreover, the performance of BL-based ACs, will be compared with previously produced ACs from RS- and SCB pulp fibers [37]. The choice of SCB for comparison is based on this residue already used in local paper mills for production of paper products.

- The importance of the adsorbate chosen should be addressed. Why MB is important to be adsorbed? Why MB constitute a problem?

Answer

The Standard tests to evaluate the adsorption capacity of any carbon material are focused on studying its surface area, as well as its adsorption capacity to methylene blue and iodine value. These tests were conducted as an indicator to the efficiency of material to remove the contaminants from liquids, especially dys. These evaluations are reported in many literatures. Moreover these tests were performed till we can compare the obtained results with the previous performed ACs from RS and SCB pulps, which evaluated by the same tests. With thanks for this inquire.

- Did the authors considered the circular economy approach? It would be interesting to link this type of work to that approach, since it deals with wastes and giving them an added-value.

Answer

Thank you for this comment; I know it will be very useful to perform the feasibility study, as a stage for production in industrial scale. But her I refereed to economic production, based on minimizing the stages via using the black liquor as a whole (without passing for precipitation to separate the lignin,... etc), moreover preserving the cost of pretreatment of water before its discharge to the drainage, for keeping the level of COD and BOD.

LORD willing, I'll perform the feasibility study after finishing from other approaches undertaken in forthcoming 3 articles, to recommend the efficient process.

3. Comments related to Experimental section

- Figure should be numbered and mentioned in the text.

Answer

Thank you, I referred to this Fig as Diagram 1.

- Page 4, Line 48: Avoid the term "our".
- Page 4, Line 50: BLs were already defined. Once again, the issue of the abbreviations should be carefully revised in the whole manuscript!

Answer

In Materials section I referred to black liquors together with the abbreviation BLs, while for sequence paragraphs and all articles I referred to these precursors by abbreviation, with thanks.

- Page 6, Line 23: Which are the reasons for that high concentration levels?

Answer

Thank you, as I illustrated before all performed tests are proceeds based on the tests which already performed in our previous articles, and till give me the opportunity to specify the efficiency of the recent activated carbon when compared with ACs produced from pulp fibers.

- Page 6, Line 29: How the authors performed the UV-vis measures? Was the matrix of the AC taken into account? The measures should be performed against the matrix. Please comment.

Answer

In the revised Ms. I added the sentences "and using pre-plot standard curve".

- Page 7, Line 49: Do the authors think that the kinetic models used may help in gathering knowledge about mechanisms of adsorption?

Answer

The fit of observed date to Pseudo-second-order kinetic model will increases the probability of the chemisorption adsorption mechanism. Moreover, fitting of experimental data to pseudo second order model and Langmuir isotherm model usually signifies chemisorption. This opinion was supported by many researchers. With many thanks for this comment.

- Page 8, Line 19: Do the authors believe that the FTIR-KBr is the best option? Would not be better to use ATR-FTIR? Please comment.

Answer

I agree about the Fourier Transform Infra-Red-Attenuated Total Reflectance (FTIR-ATR), is best technique, because it can distinguish components (macerals) of carbon material which have diverse chemical compositions and physical properties, quantifying the abundance of chemical functional groups, but this Instrument not available in many Institutes like us. Therefore, limited studies have reported via using ATR-FTIR spectra for qualitatively evaluating ACs or other carbon materials. Moreover, we focused on instruments available to us in my Institute, and all tests covered by private money, till my student have publ's. for her promotion. The FTIR spectra only show the presence of surface functional groups.

Could you please check the item **Funding which reported before References section** "Funding is self-employed from all authors".

- Page 8, Line 29: Correct the title of section 2.3.2.4 to "microscopy".

Answer

We corrected this word, with thanks.

4. Comments related to Results section

- I wonder why the elemental analysis and the TGs were performed solely for the BLs and not for the ACs. I think it would be useful to do such characterization for the ACs as well. Please comment.

Answer

Many thanks for this comment, but could you please recheck the paragraphs related to these characteristics. These studies were concerned on examine the role of pulping process on performance of black liquor as a precursor for production of ACs. Therefore the indicator factor for changing the adsorption capacity of the resulted ACs, is chemical constituents of BL vs pulping process, The chemical analyses test were carried out to indicate the constituents of black liquors (lignin and hydrolyzed hemicelluloses), which are related to delignification degree of pulping process. The H/C and O/C were calculated as supporting

data to pulping process. I think these constituents will further affect on thermal stability, as all knowledge about the greater stability of lignin than hemicellulose.

The TGA is carried out to indicate the trend of weight loss of BL versus carbonization temperature. The pulping processes significant affect on weight loss, and consequently the char (weight remain). I think this analyses on precursors will be very useful than on produced ACs.

This study also illustrated the efficiency of BLs than pulp fibers, where the weight loss of un- and RS and SCB pulps are greater than the case of BLs. This is attributed to the higher organic matter contents, especially cellulose (C and H) of un- and pulped fibers as compared with BLs. We ascribed this observation to the degradation of cellulose leads to formation of levoglucosan which promotes the volatilization stage. This difference is greater in case of soda pulping of RS.

- Do the authors have the possibility of measure the TOC of the materials? It would be an important measure to complement the characterization.

Answer

Thank you, I think the TGA analysis of BLs will illustrates the inorganic materials included the carbon residues after pyrolysis, at temperature in the range 500-900 °C. The remained weight after pyrolysis will give us information dealing carbon material purity

The silica content included the carbon material, may provides improvement in adsorption capacity, therefore we focused on applying the ACs as a whole.

LORD willing we will take this measurement in our further publication.

5. Comments related to Tables and Figures

- Figure 3 is a duplication of data from Table 3. Use either the Figure or the Table.

Answer

Thank you, but could you please check again Fig. 3, as it included the data of S_{BET} and V_T versus BLs-ACs, in comparison with previous data related to ACs from pulp fibers; While Table 3 only included all textural characterization of various BL-ACs

- Kinetics and isotherms should be presented in a figure.

Answer

We employed kinetics and isotherms, related to AC from RS-BL, as example, because it provided well-defined adsorption (please see page 22).

Based on the shift in number of Figs, as 3rd comment of Reviewer 2 comments (***3.The structure of this paper should well organize. For example, the IR and SEM results should be in front of the adsorption results of iodine and MB***), this Fig. will take the No. Fig. 7.

I'm sorry about the language; we carefully corrected the language in revised version, with thanks.

Reviewer: 2

This manuscript presents a simple procedure to prepare cost-effective porous activated carbons from black liquor during production of paper pulp using phosphoric acid as activation agent. The result carbons were used as aqueous adsorbates. The work is interesting and the manuscript is well organized. However, the authors should address the questions below:

1. As the author said, there are lot of micropores for the ACs, so for the N_2 adsorption-desorption isotherms, more points at the low pressure range (<0,1) should be present.

Answer

Thank you about your decision which give us the opportunity to extent our publication in Royal Society Open Science.

With regard to reviewer comment, the t-plot or alpha -plot is used to estimate the type of porosity (micro/or meso), but because the Instrument used in this test (USA Nova 3200 Nitrogen Phesosorption Quantachrom Instrument), not in mine and the test are performed in other Lab by Technician, therefore I can't suggest and put many points to measure, which requires time *consuming*, especially the liquid nitrogen is not available for this Lab all times, May we wait months to analyze the samples due to this problem. The measurements of micro, meso and total are already performed by software of Instrument.

Based on our information's' on applying SPE method, despite no data appear at low pressure, the test indicates the micropore volume included the sample. *therefore please forgive me if I can't response*

2. For confirming, the pore size distribution from N₂ adsorption-desorption isotherms should also add in the text.

Answer

The paragraph under sub.title **3.2.1. Surface textural characterization**, was modified

3. The structure of this paper should well organize. For example, the IR and SEM results should be in front of the adsorption results of iodine and MB.

Answer

We performed the changes in sub-titles of Experimental and Results and discussion sections, with thanks.

4. There are lot of recent works about ACs for iodine and MB adsorption, so some new reference should add.

5. A new part about the comparison should also present.

Answer

Please check the revised Ms., page # 13, under sub.title **3.2.4** , 2nd paragraph, Also Table 6 which was added for comparison. References 57-60 were added.

6. The Conclusions likes an experiment summary rather a Conclusion which should give not only the main conclusion but also echoing the introduction.

Answer

We modified the conclusion; I hope it will be convenient with reviewer request.

7. The quality of “Scheme of the Experimental work” is too low. It happened also for all the equations in this paper.

Answer

We improved the quality of both diagram 1 and equations. May the bad equations quality in original submission due to I converted the file from word 2007 to 2003. Please forgive me about it, with thanks

Reviewer: 3

Comments to the Author(s)

Dear Authors

Wonderful work have been done by you, and check any residue found after the Activated carbon production by is the black liquor.

Thank you very much for your kind recommendation which will give us the opportunity to publish in Roy Soc. Open Sci.

Finally, thanks again for all efforts by editorial staff and reviewers, and we hope (LORD willing) extent our publication in your Journal and our corrections meet with approval.

v. best, may our LORD bless you about your time
Altaf H. Basta (Res.Prof.)
NRC, Cairo, Egypt